# Learning Transferable Features for Point Cloud Detection via 3D Contrastive Co-training

**Yihan Zeng**[1]* **Chunwei Wang**[1]* **Yunbo Wang**[1]
**Hang Xu**[2] **Chaoqiang Ye**[2] **Zhen Yang**[2] **Chao Ma**[1]†
[1] MoE Key Lab of Artificial Intelligence, AI Institute, Shanghai Jiao Tong University
[2] Huawei Noah's Ark Lab
{zengyihan,weiwei0224,yunbow,chaoma}@sjtu.edu.cn
{xu.hang,yechaoqiang,yang.zhen}@huawei.com

## Abstract

Most existing point cloud detection models require large-scale, densely annotated datasets. They typically underperform in domain adaptation settings, due to geometry shifts caused by different physical environments or LiDAR sensor configurations. Therefore, it is challenging but valuable to learn transferable features between a labeled source domain and a novel target domain, without any access to target labels. To tackle this problem, we introduce the framework of **3D Contrastive Co**-training (**3D-CoCo**) with two technical contributions. First, 3D-CoCo is inspired by our observation that the bird-eye-view (BEV) features are more transferable than low-level geometry features. We thus propose a new co-training architecture that includes separate 3D encoders with domain-specific parameters, as well as a BEV transformation module for learning domain-invariant features. Second, 3D-CoCo extends the approach of *contrastive instance alignment* to point cloud detection, whose performance was largely hindered by the mismatch between the fictitious distribution of BEV features, induced by pseudo-labels, and the true distribution. The mismatch is greatly reduced by 3D-CoCo with transformed point clouds, which are carefully designed by considering specific geometry priors. We construct new domain adaptation benchmarks using three large-scale 3D datasets. Experimental results show that our proposed 3D-CoCo effectively closes the domain gap and outperforms the state-of-the-art methods by large margins.

## 1 Introduction

3D point cloud detection shows remarkable significance in real-world scenarios, such as autonomous driving [14, 43, 38, 44], in which the recent progress is largely driven by the emergence of high-precision LiDAR sensors and large-scale, densely annotated point cloud datasets [1, 6, 26]. Most existing 3D detection models assume that the training domain and testing domain are independently and identically distributed. In practice, however, domain shifts are often inevitable due to differences in physical environments or LiDAR sensor configurations, including different numbers of laser beams and installation positions, etc. To address this issue, we present an early study of the unsupervised domain adaptation problem for point cloud detection, which aims to effectively adapt 3D detectors from the labeled source domain to a novel unlabeled target domain by learning transferable features.

Previous domain adaptation approaches for image data [27, 12, 24, 2] are not readily applicable to point clouds. As shown in Fig. 1, different from the domain shifts of 2D scenes that usually

---

* The first two authors have equal contributions.

† C. Ma is the corresponding author.

35th Conference on Neural Information Processing Systems (NeurIPS 2021).

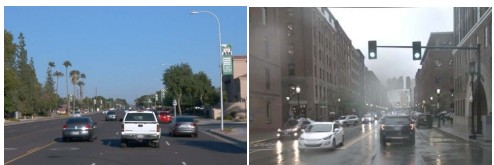 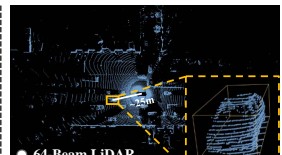 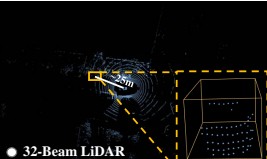

Figure 1: Comparisons of domain shifts between 2D and 3D scenes. **Left:** Domain shifts in 2D scenes are mainly reflected in appearance variations, *e.g.*, weather or environment changes in autonomous driving. **Right:** 3D domain shifts are generally represented as geometry variations, which not only arise from external environments, but also come from the internal sensor configurations.

exists in the image appearance, those of 3D scenes are mainly reflected in the geometry variations of point clouds. Since 2D images from different domains have the same grid topology of uniformly distributed pixels, most of the domain adaptation methods exploit image encoders with domain-sharing parameters, which is also adopted by existing 3D transfer learning models such as PointDAN [20]. However, due to the severe low-level geometry shifts between different point sets, we believe that some features are transferable while some features are not on 3D object detection, which requires a rethink of the transferability of different levels of the point cloud representations. To this end, we propose a novel framework named 3D Contrastive Co-training (3D-CoCo), whose architecture contains separate 3D encoders with domain-specific parameters, a domain-agnostic BEV transformation module, and the final detection head. The key idea of the architecture design is that the BEV features with similar grid structures to images can be more transferable than low-level 3D features, so that they can be better integrated with advanced transfer learning techniques in 2D vision and thus greatly reduce the geometry shift. Another benefit of the co-training architecture with domain-specific encoders is that in addition to improving domain adaptation results, it also maintains the in-domain performance.

Furthermore, 3D-CoCo is also featured by a new end-to-end contrastive learning framework, which contains two main components, *i.e.*, contrastive instance alignment based on bird-eye-view (BEV) features and hard sample mining with transformed point clouds. The insight of the contrastive instance alignment is to push the feature centroids of similar sample clusters, induced by pseudo-labels, closer to each other, no matter they are in the same domain or different domains. Besides, we consider the mismatch between the true distribution of BEV features and the fictitious one used for contrastive learning, which is caused by biased pseudo-labels. Specifically, we take advantage of the editability of point clouds and perform hard sample mining by applying specific transformation functions to 3D data. The hard samples, as an effective supplement to contrastive co-training, can further reduce the geometry shift across domains and prevent the adaptation model from falling into the local minima.

Notably, there exists another line of work discussing transfer learning for point cloud detection [28, 35, 3], which utilizes the *self-training* pipeline that retrains the model with pseudo-labels on target data. Compared to these approaches, we adopt a different problem setup, using labeled source domain data and unlabeled target domain data for *co-training*. Through ablation studies, we observe that the separate encoders and the contrastive co-training scheme can progressively filter domain-specific features and learn more transferable knowledge across domains.

We evaluate the effectiveness and generalizability of 3D-CoCo across three widely-used autonomous driving datasets collected by heterogeneous LiDAR sensors, including Waymo [26], nuScenes [1], and KITTI [6]. 3D-CoCo is shown to significantly outperform existing approaches for point cloud detection on different unsupervised domain adaptation benchmarks.

## 2 Preliminaries

**Problem setup.** The traditional setup of point cloud detection is to learn a base 3D object detector $D$ that classifies and localizes $m$ objects represented by $Y$ from a point cloud $P$:

$$D : P \to Y, \tag{1}$$

where $P = \{p_1, \ldots, p_n\}$ consists of $n$ points $p_i \in \mathbb{R}^d$ scattered over the 3D coordinate space. The dimension $d$ is set to 4, including the coordinate $(x, y, z)$ and an additional intensity $i$. The annotations $Y = \{y_1, \ldots, y_m\}$ consists of a set of class labels and bounding box coordinates,

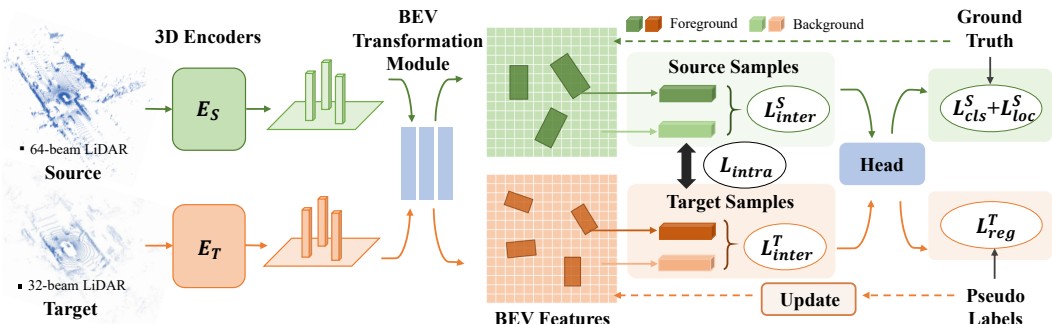

Figure 2: An illustration of the proposed 3D-CoCo model, which contains domain-specific 3D encoders and performs contrastive adaptation on BEV features for instance-level feature alignment.

which is represented by the size $(l, w, h)$, center $(c_x, c_y, c_z)$, and heading angle $r$ of each 3D object. Typically, $D$ is trained in a supervised manner by minimizing the classification and localization errors between the ground truth $Y$ and the prediction $\widehat{Y}$ over the training (source) dataset. In this paper, we specifically study the problem of *unsupervised domain adaptation* for point cloud detection, which aims to adapt the detector $D_\theta$ parametrized by $\theta$ from a labeled source domain $\{(P_i^S, Y_i^S)\}_{i=1}^{N_S}$ to an unlabeled target domain $\{P_i^T\}_{i=1}^{N_T}$, where $N_S$ and $N_T$ are the numbers of training samples:

$$D_\theta : P^S \cup P^T \to Y^S \cup Y^T. \tag{2}$$

Compared to the typical setup of point cloud detection, we focus more on the performance of the model on the test set of the target domain, which requires additional well-designed modules for learning transferable features.

**Modern point cloud detectors.** Modern 3D point cloud detectors [38, 43, 14] usually consist of three modules: a 3D encoder $E$, a bird-eye-view (BEV) transformation module $U$, and a detection head $H$. The 3D encoder quantizes the point cloud into regular grids for feature extraction. The BEV transformation module, usually in forms of 2D convolutional layers, produces fixed-size BEV feature maps $M \in \mathbb{R}^{W \times L \times F}$, where width $W$ and length $L$ are based on the resolution of bins, and $F$ denotes the channel number. The detection head takes $M$ as inputs and produces detection results $\widehat{Y}$. To demonstrate the generality of the proposed method, we adopt two mainstream architectures, *i.e.*, VoxelNet [43] and PointPillars [14], with different point cloud processing pipelines as alternative options of the 3D encoder. VoxelNet quantizes the point cloud into small 3D voxel features and then uses a 3D CNN to compress them into 2D BEV space along the height of the voxel, while PointPillars quantizes the point cloud into vertical pillars on fixed-size 2D grids and then performs linear transformation and max-pooling on each pillar to obtain the BEV representation. Neither of them explicitly considers the domain shift in transfer learning setups.

## 3 Method

We present 3D-CoCo as a feasible solution to the unsupervised domain adaptation task in point cloud detection. It has two contributions to learning transferable features from heterogeneous geometries, which respectively reside in the new architecture design, as shown in Fig. 2, and the framework of contrastive instance alignment enhanced by hard sample mining, as shown in Alg. 1 and Fig. 3.

### 3.1 3D-CoCo Architecture

**Domain-specific 3D encoders.** Transfer learning in 3D scenes may suffer from dramatic geometry shifts, such as density variations and different occlusion ratios of point clouds, due to different physical environments and sensor configurations. Although some work has explored model pretraining on 3D pretext tasks [41, 30], in contrast with 2D scenarios, 3D vision still lacks a transferable, well-pretrained backbone. One possible reason is that it is very difficult to reduce the domain shift in geometric representations at the bottom of the 3D encoder. Intuitively, we expect the 3D detection network to progressively process domain-specific non-transferable features and learn domain-invariant semantic features. As shown in Fig. 2, we present a novel model architecture with domain-specific 3D encoders, which learn different mapping functions to parse and convert LiDAR points into the

**Algorithm 1:** The learning procedure of 3D contrastive co-training (3D-CoCo)

---

**Input:** The labeled point set from the source domain $\{(P_i^S, Y_i^S)\}_{i=1}^{N_S}$, the unlabeled point set from the target domain $\{P_i^T\}_{i=1}^{N_T}$, the maximum number of update stages $K$

**Output:** Learned network weights $\theta$

1   $D' \leftarrow \{(P_i^S, Y_i^S)\}_{i=1}^{N_S}$              $\triangleright$ Pretrain the base detector according to Eq. (3)

2   $\{\overline{Y}_i^T\}_{i=1}^{N_T} \leftarrow (D', \{P_i^T\}_{i=1}^{N_T})$      $\triangleright$ Generate pseudo-labels for target domain samples

3   $(\{(P_i^T, \overline{Y}_i^T)\}_{i=1}^{N_T}) \leftarrow \text{HSM}(\{(P_i^T, \overline{Y}_i^T)\}_{i=1}^{N_T})$    $\triangleright$ Mine hard samples to augment the target set

4   $D_\theta \leftarrow D'$                       $\triangleright$ Initialize the model with the base detector

5   $D_\theta^0 \leftarrow (\{(P_i^S, Y_i^S)\}_{i=1}^{N_S}, \{(P_i^T, \overline{Y}_i^T)\}_{i=1}^{N_T})$      $\triangleright$ Warm-up according to Eq. (6)

6   **for** $k = 1, \ldots, K$ **do**

7      $\{\overline{Y}_i^T\}_{i=1}^{N_T} \leftarrow (D_\theta^{k-1}, \{P_i^T\}_{i=1}^{N_T})$         $\triangleright$ Update pseudo-labels

8      $(\{(P_i^T, \overline{Y}_i^T)\}_{i=1}^{N_T}) \leftarrow \text{HSM}(\{(P_i^T, \overline{Y}_i^T)\}_{i=1}^{N_T})$   $\triangleright$ Append new hard samples to the target set

9      $D_\theta^k \leftarrow (\{(P_i^S, Y_i^S)\}_{i=1}^{N_S}, \{(P_i^T, \overline{Y}_i^T)\}_{i=1}^{N_T})$     $\triangleright$ Model update according to Eq. (6)

10     $D_\theta^{k-1} \leftarrow D_\theta^k$

---

bird-eye-view (BEV) space for different domains. It is worth noting that the co-training architecture not only benefits the adaptation performance on the target domain but also contributes to maintaining the performance on the source domain, in the sense that learning transferable features upon separate encoders can facilitate bi-directional knowledge sharing.

**Domain-agnostic BEV transformation module.** The 2D transformation module is co-trained with data samples from both source and target domain. It further compresses the outputs of the domain-specific 3D encoders into BEV feature maps $M$. The BEV features are supposed to be more transferable, because they have similar structures to the grid-based feature maps in 2D vision and can therefore be easily integrated into off-the-shelf transfer learning techniques. Based on $M$, we perform the contrastive alignment training scheme to encourage the learning of domain-invariant features.

**Detection head.** The detection head classifies and localizes 3D objects from the BEV feature maps $M$. Given a labeled sample from the source domain, the detection head is trained to minimize

$$\mathcal{L}_{\text{det}} = \mathcal{L}_{\text{cls}}^S + \mathcal{L}_{\text{loc}}^S, \tag{3}$$

where $\mathcal{L}_{\text{cls}}^S$ and $\mathcal{L}_{\text{loc}}^S$ respectively indicates the classification and localization error. For an unlabeled sample from the target domain, we use a regularization of the localization error $\mathcal{L}_{\text{reg}}^T$ between the predictions and the pseudo-labels, which encourages a rapid adaptation to the target domain.

## 3.2 Hard Sample Augmented Contrastive Alignment

Since the features of point clouds are sparsely distributed, it is difficult to achieve effective matching between domains by using global distribution alignment. We thus propose to exploit fine-grained alignment at the instance level, which is enhanced by hard sample mining to avoid bad local minima.

**Foreground and background proposals.** We first construct foreground and background instance-level features for contrastive domain alignment, respectively. According to the predicted pseudo-labels $\overline{Y}^T = \{\overline{y}_1^T, \ldots, \overline{y}_m^T\}$, we obtain foreground detection proposals on the BEV features. We then apply a keypoint-based feature extraction approach to each proposal. Concretely, we assign $R \times R \times 1$ equally distributed keypoints to the proposal box, and then learn the feature of each keypoint by performing bilinear interpolation on the feature map and get the average-pooling value as the final representation. For background samples, we randomly select grid features from areas outside the proposal boxes on the feature map. It is worth noting that the proportion of blank areas without any points in the background region is relatively large, while the proportion of non-blank areas with background points is relatively small. There is a gap between the learned features of these two kinds of background samples. To alleviate the sample imbalance, we equally select a certain number of features from these two groups of background samples.

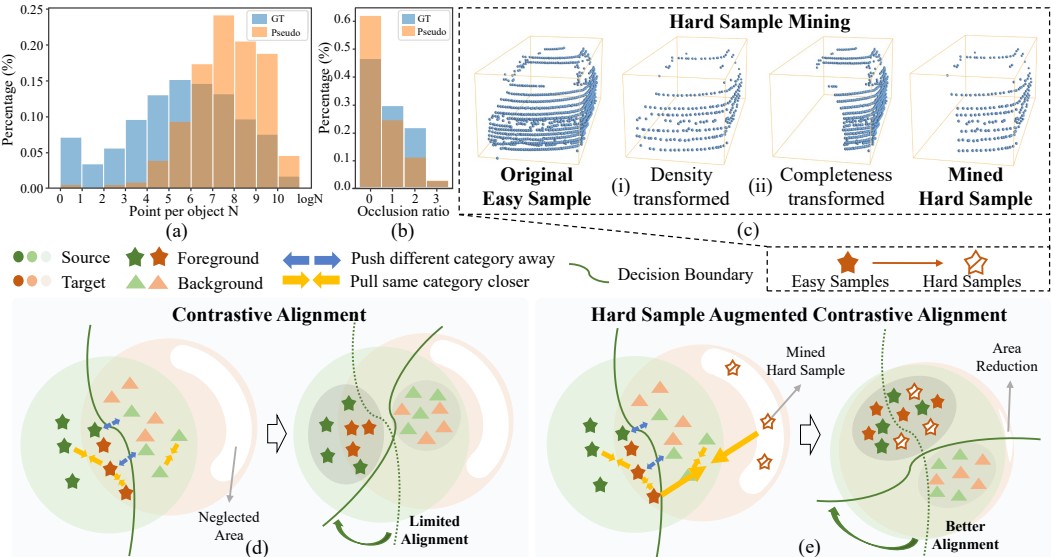

Figure 3: The idea of the hard sample augmented contrastive alignment. **(a-b)** A simple use of contrastive alignment introduces the mismatch in point density and occlusion ratio between sample distributions of pseudo-labels and ground truths in the target domain. **(c)** Hard sample mining transforms point clouds by considering the specific geometry mismatches. **(d)** The original contrastive alignment focuses more on the alignment of easy samples in 3D scenes rather than the easily neglected hard samples with severe occlusions or density variations. **(e)** The contrastive alignment scheme of 3D-CoCo is effectively augmented by the transformed hard samples to achieve further alignment.

**Contrastive instance alignment induced by pseudo-labels.** The core idea of contrastive alignment is to drive the feature centroids of similar samples closer between domains. Specifically, we choose the positive instance sample pair $(I_i^S, I_j^T)$ based on a similarity-priority criterion as follows:

$$j^\star = \mathbf{P}^c(i) = \arg\max_{1 \le j \le N_c^T} \{\Phi(I_i^S, I_j^T)\}, \ \ 1 \le i \le N_c^S, \ \ c = 1, 2, \ldots, |C|, \tag{4}$$

where $N_c^S$ denotes the total number of source samples in category $c$ and $|C|$ denotes the total number of categories. For each $I_i^S$, it finds a positive sample $I_{j^\star}^T$ from the target domain under the same category $c$, where $\Phi(\mathbf{x}, \mathbf{y}) = \frac{\mathbf{x} \cdot \mathbf{y}}{\|\mathbf{x}\|\|\mathbf{y}\|}$ calculates the cosine similarity between features of a source sample and a target candidate within the same category. In addition to minimizing the intra-class distance of the same category between domains, we also constrain the inter-class distance between different categories within the same domain. The objective of contrastive alignment is formulated as

$$\mathcal{L}_{\text{intra-class}}(S, T) = -\sum_{c=1}^{|C|} \sum_{i \in N_c^S} \log \frac{\exp(I_i^S \cdot I_{j^\star}^T / \tau)}{\exp(I_i^S \cdot I_{j^\star}^T / \tau) + \sum_{j \in N_{|C| \backslash c}^T} \exp(I_i^S \cdot I_j^T)},$$

$$\mathcal{L}_{\text{inter-class}}(S) = -\sum_{c=1}^{|C|} \sum_{i \in N_c^S, j' \in N_c^S, i \ne j'} \log \frac{\exp(I_i^S \cdot I_{j'}^S / \tau)}{\exp(I_i^S \cdot I_{j'}^S / \tau) + \sum_{j \in N_{|C| \backslash c}^S} \exp(I_i^S \cdot I_j^S)}, \tag{5}$$

$$\mathcal{L}_{\text{inter-class}}(T) = -\sum_{c=1}^{|C|} \sum_{i \in N_c^T, j' \in N_c^T, i \ne j'} \log \frac{\exp(I_i^T \cdot I_{j'}^T / \tau)}{\exp(I_i^T \cdot I_{j'}^T / \tau) + \sum_{j \in N_{|C| \backslash c}^T} \exp(I_i^T \cdot I_j^T)},$$

$$\mathcal{L}_{\text{uda}} = \mathcal{L}_{\text{intra-class}}(S, T) + \mathcal{L}_{\text{inter-class}}(S) + \mathcal{L}_{\text{inter-class}}(T),$$

where $\tau$ denotes a temperature hyper-parameter, which is set to 0.07 in our experiments. In the total adaptation loss $\mathcal{L}_{\text{uda}}$, all pairwise relations of samples between domains and within domains are considered, which enhances both the intra-class transferability as well as the inter-class discriminability. Finally, by combining the loss terms in Eq. (3), Eq. (5), and the regularization term for target domain $\mathcal{L}_{\text{reg}}^T$, we optimize the model $D_\theta$ by

$$\arg\min_\theta \ \mathcal{L}_{\text{det}} + \lambda \mathcal{L}_{\text{uda}} + \mathcal{L}_{\text{reg}}^T, \tag{6}$$

where $\lambda$ is the hyper-parameter for the domain adaptation term, set as 0.5 in experiments.

**Transformed point clouds as hard samples.** A straightforward use of the contrastive instance alignment tends to introduce the mismatch between the sample distributions obtained by pseudo-labels and ground truths on the target domain. First, as shown in Fig. 3(a), pseudo-labels are more concentrated in the patterns with dense point clouds than those with sparse point clouds. Second, as shown in Fig. 3(b), pseudo-labels cannot completely cover the patterns of severe occlusions. Therefore, most instances induced by positive pseudo-labels can be viewed as "easy samples" with sufficient points or complete geometry. However, we believe that the neglected "hard samples", which are more likely to be distributed in the marginal area shown in Fig. 3(d), are equally important to 3D transfer learning. As shown in Fig. 3(e), mining the hard samples can further promote distribution alignment and prevent the model from overfitting bad local minima. The key to creating fictitious hard samples is to consider the priors of geometry variations indicated by Fig. 3(a-b). We here propose two mechanisms to create fictitious hard samples. As shown in Fig. 3(c), the first transformation method uniformly discards the points from existing dense point clouds, which simulates the change of the number of laser beams. The second method simulates object occlusions by breaking the complete geometry of easy samples. Concretely, we calculate the viewpoint of a certain sample, randomly select a part of the viewpoint, and discard the point cloud on these angles. Compared to common augmentation strategies such as rotation and flipping that were previously applied to 3D detection [38, 32], the transformed point clouds focus on effective contrastive instance alignment by reducing the distribution mismatch of the target domain induced by pseudo-labels, rather than aiming at enriching sample diversity of the source domain.

**Overall training procedure.** We propose a step-wise training procedure with a warm-up process as shown in Alg. 1. Specifically, we first pretrain a source detector on the labeled source domain and use it to generate pseudo-labels on the target set. We then conduct hard sample mining (HSM) and augment the target set. Next, we warm up the 3D-CoCo detection model following Eq. (6), which allows a more stable convergence in the early stages of training. For the remaining epochs, we update the pseudo-labels using the ensembling and voting mechanism [35]. Throughout step-wise co-training, $D_\theta$ gradually adapts to the target domain while maintaining the in-domain performance.

## 4 Experiments

### 4.1 Experimental Setup

**Datasets.** We evaluate 3D-CoCo on three widely used LiDAR-based datasets, including Waymo [26], nuScenes [1], and KITTI [6]. Each dataset has specific properties in external environments (*i.e.*, traffic condition) and internal sensor configurations (*i.e.*, number of beams), so there exist huge domain gaps between them. Specifically, Waymo is collected in the United States with multiple weather conditions throughout the day using 5 LiDAR sensors. The nuScenes dataset is collected in the United States and Singapore by a 32-beam LiDAR sensor. KITTI is collected in Germany in sunny daytime by a 64-beam LiDAR sensor. We construct 4 domains adaptation benchmarks between datasets including: (i) Waymo→nuScenes, (ii) nuScenes→Waymo, (iii) Waymo→KITTI, and (iv) nuScenes→KITTI. We use the common category of the three datasets, *i.e.*, Car/Vehicle. Here, KITTI is only used as the target domain as it is much smaller than the other two datasets.

**Compared models.** As shown in Table 1, 3D-CoCo is firstly compared with the "Source Only" model, which is trained with only source domain data. We include two existing approaches for cross-domain 3D detection: SN [28] normalizes the object size of the source domain by leveraging the object-level statistics of the target domain. ST3D [35] is a self-training pipeline that achieves the state-of-the-art domain adaptation results by using pseudo-labels of target data for retraining. We re-implement SN and ST3D on the same base detector as ours. Finally, 3D-CoCo is also compared with the "Oracle" model, which is trained with labeled target domain data, to roughly represent the optimal performance of an adaptation model on the target domain.

**Evaluation metrics.** We use the average precision (AP) as the evaluation metric for both BEV IoUs and 3D IoUs under an IoU threshold of 0.7. We also adopt the domain adaptation metric named Closed Gap from Yang *et al.* [35], which is defined as $\frac{AP_{model} - AP_{source}}{AP_{oracle} - AP_{source}} \times 100\%$, the higher, the better.

**Implementation details.** We follow Yin *et al.* [38] to build the base detector with two alternative 3D encoders including VoxelNet and PointPillars. We set the voxel size to $(0.1, 0.1, 0.15)m$ for

| Task | Method | VoxelNet | | | | PointPillars | | | |
|---|---|---|---|---|---|---|---|---|---|
| | | $AP_{BEV}$ | Closed Gap | $AP_{3D}$ | Closed Gap | $AP_{BEV}$ | Closed Gap | $AP_{3D}$ | Closed Gap |
| N → K | Source Only | 46.7 | - | 11.1 | - | 22.8 | - | 0.5 | - |
| | SN [28] | 35.4 | −30.13% | 22.9 | +19.41% | 39.3 | +26.61% | 2.0 | +2.11% |
| | ST3D [35] | 54.4 | +20.53% | 38.3 | +44.74% | 60.4 | +60.65% | 11.1 | +14.91% |
| | 3D-CoCo | **77.1** | **+81.07%** | **65.6** | **+89.64%** | **77.0** | **+87.42%** | **47.2** | **+65.68%** |
| | Oracle | 84.2 | - | 71.9 | - | 84.8 | - | 71.6 | - |
| W → K | Source Only | 55.1 | - | 18.9 | - | 47.8 | - | 11.5 | - |
| | SN [28] | 50.6 | −15.46% | 36.7 | +33.58% | 27.4 | −55.14% | 6.4 | −8.49% |
| | ST3D [35] | 72.6 | +60.14% | 54.8 | +67.74% | 58.1 | +27.84% | 23.2 | +19.47% |
| | 3D-CoCo | **77.0** | **+75.26%** | **61.9** | **+81.13%** | **76.1** | **+76.49%** | **42.9** | **+52.25%** |
| | Oracle | 84.2 | - | 71.9 | - | 84.8 | - | 71.6 | - |
| W → N | Source Only | 32.4 | - | 21.1 | - | 27.8 | - | 12.1 | - |
| | ST3D [35] | 32.8 | +2.34% | 21.1 | +0.00% | 30.6 | +13.21% | 15.6 | +18.23% |
| | 3D-CoCo | **39.2** | **+39.77%** | **25.4** | **+37.07%** | **33.1** | **+25.00%** | **20.7** | **+44.79%** |
| | Oracle | 49.5 | - | 32.7 | - | 49.0 | - | 31.3 | - |
| N → W | Source Only | 40.2 | - | 22.9 | - | 28.1 | - | 8.5 | - |
| | ST3D [35] | 42.7 | +6.46% | 29.0 | +11.78% | 35.1 | +15.63% | 14.0 | +13.13% |
| | 3D-CoCo | **56.7** | **+42.64%** | **33.0** | **+19.50%** | **50.3** | **+49.55%** | **29.5** | **+50.12%** |
| | Oracle | 78.9 | - | 74.7 | - | 72.9 | - | 50.4 | - |

Table 1: Results in average precision and the corresponding closed gaps for unsupervised domain adaptation. Please see the text for the definition of the metric. **N:** nuScenes; **K:** KITTI; **W:** Waymo.

VoxelNet and $(0.1, 0.1)m$ for PointPillars. We use the Adam optimizer [13] with a learning rate of $1.5 \times 10^{-3}$. We set the maximum number of training epochs to 30 for KITTI and 20 for Waymo and nuScenes, with a warm-up process taking half of the total epochs. For pseudo-labels generation, we apply a high-pass threshold of 0.7 to IoU to obtain foreground samples, and a low-pass threshold of 0.2 for background samples. To reduce the domain shift of object size between datasets [28], we use the random object scaling (ROS) strategy [35] with a scaling factor in the range of $[0.75, 0.9]$ when adapting the model to KITTI. In this way, different from Statistical Normalization (SN) [28], our approach does not require accurate prior knowledge of the target domain statistics.

## 4.2 Main Results

As shown in Table 1, 3D-CoCo outperforms all compared models by large margins on all adaptation benchmarks. Especially on nuScenes→KITTI and Waymo→KITTI based on the VoxelNet backbone, 3D-CoCo closes the domain gap in $AP_{3D}$ by around $81\% \sim 89\%$. Furthermore, for the adaptation tasks between the two large-scale datasets, *i.e.*, Waymo and nuScenes, 3D-CoCo also achieves considerable improvement, closing the domain gap in $AP_{3D}$ by $37\%$ on VoxelNet and $50\%$ on PointPillars. Notably, despite taking 3D domain shift into account, SN and ST3D achieve relatively small improvements under the domain adaptation setups, or even have a negative effect on the base model (source only). In contrast, although starting with low-quality pseudo labels, 3D-CoCo still performs well due to the effective co-training that incorporates labeled source data and augmented hard samples. The overall results validate the transferability of 3D-CoCo on different unsupervised domain adaptation benchmarks, and its ability to generalize to different detection networks.

## 4.3 Ablation Studies

**Architecture designs.** All ablation studies are conducted on nuScenes→KITTI, using VoxelNet as the network backbone. First, Table 2(I) compares the results of using different parameter-sharing strategies in the model architecture. By replacing the domain-specific 3D encoders of 3D-CoCo with

| | Method | Target | | Source | | | Method | Target | | Source | |
|---|---|---|---|---|---|---|---|---|---|---|---|
| | | $AP_{BEV}$ | $AP_{3D}$ | $AP_{BEV}$ | $AP_{3D}$ | | | $AP_{BEV}$ | $AP_{3D}$ | $AP_{BEV}$ | $AP_{3D}$ |
| (I) | Source Only | 45.2 | 32.6 | **32.2** | 19.4 | (II) | Source Only | 51.4 | 32.2 | 44.0 | 28.0 |
| | $1E + 1U$ | 74.0 | 59.1 | 24.5 | 14.3 | | $1E + 1U$ | 73.5 | 53.7 | 39.1 | 25.5 |
| | $2E + 2U$ | 77.5 | 60.3 | 31.6 | 18.1 | | $2E + 2U$ | 64.0 | 26.0 | 43.5 | 23.5 |
| | $2E + 1U$ (Ours) | **77.1** | **65.6** | 30.6 | **19.6** | | $2E + 1U$ (Ours) | **77.1** | **55.6** | **44.6** | **30.0** |

Table 2: Ablations on the architecture design. $1E$ and $2E$ respectively denotes using a domain-sharing 3D encoder and separate domain-specific encoders. $1U/2U$ denotes using shared/separate BEV transformation module(s). All models are trained in the proposed contrastive alignment framework with the random object scaling technique. In (I), the ROS scaling factor ranges in $[0.75, 0.9]$; In (II), it ranges in $[0.75, 1.1]$. We report both in-domain and cross-domain performance.

| | Bkgd | Sim | Proto | $AP_{BEV}$ | $AP_{3D}$ |
|---|---|---|---|---|---|
| (a) | | | | 76.5 | 62.1 |
| (b) | ✓ | | | 76.8 | 63.3 |
| (c) | ✓ | ✓ | | **77.1** | **65.6** |
| (d) | ✓ | | ✓ | 72.7 | 59.2 |

Table 3: Ablations on contrastive alignment schemes. **Bkgd:** The balanced background sampling strategy; **Sim:** Similarity-priority criterion; **Proto:** Prototype-level alignment as opposed to the instance-level alignment.

| | Rand | Unif | Pers | $AP_{BEV}$ | $AP_{3D}$ |
|---|---|---|---|---|---|
| (e) | | | | 74.0 | 58.8 |
| (f) | ✓ | | | 74.2 | 61.3 |
| (g) | | ✓ | | 76.2 | 62.5 |
| (h) | | | ✓ | 76.0 | 61.4 |
| (i) | | ✓ | ✓ | **77.1** | **65.6** |

Table 4: Ablations on hard sample mining. **Rand:** Dropping points randomly; **Unif:** Dropping points uniformly; **Pers:** Dropping points of certain perspectives to simulate occlusions.

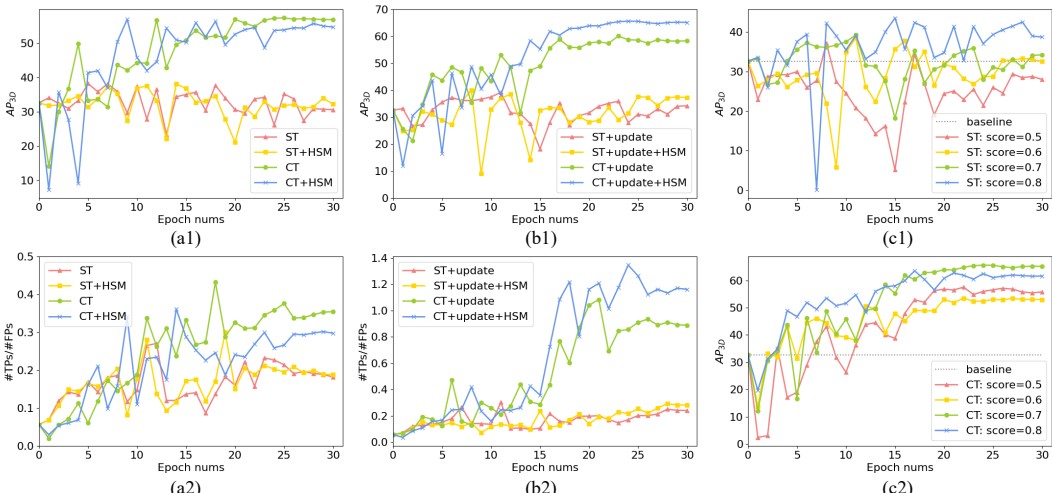

Figure 4: Comparisons of self-training **(ST)** and the proposed co-training **(CT)** methods. **(a1-a2)** Training without updating pseudo-labels. **(b1-b2)** Training with progressively updated pseudo-labels. **(c1-c2)** Training with pseudo-labels filtered by different confidence scores. $AP_{3D}$ indicates the detection accuracy. The ratio of TPs/FPs indicates the detection noise. **HSM:** Hard sample mining.

a domain-sharing encoder, we observe that the in-domain performance in $AP_{3D}$ drops by $6.5\%$ and the cross-domain performance drops by $5.3\%$, which indicates the difficulty of learning transferable features on raw point clouds due to the low-level geometry shift. We further evaluate a baseline model that contains separate BEV transformation modules, and find that the domain adaptation performance yields a $5.3\%$ drop in $AP_{3D}$. It demonstrates the transferability of the BEV features. Besides, our model can work well with ROS, which is designed to reduce the object size bias on the target domain but inevitably degrades the localization accuracy on the source domain. With different values of the ROS scaling factor, as shown in Table 2(II), our model consistently achieves the performance gain in both in-domain and cross-domain evaluation setups.

**Contrastive learning schemes.** By comparing the baseline models (a) and (b) in Table 3, we observe that the balanced background sampling strategy effectively improves the adaptation results. By further including model (c) into comparison, which is the final proposed model, we validate the effectiveness of using the similarity-priority criterion. We further compare 3D-CoCo with existing prototype-level alignment methods [18, 40] that calculate normalized features of all samples for each category as category-level prototypes for alignment. From Table 3, due to the ambiguity of the prototypes, the baseline model (d) performs much worse than the proposed model (c) with instance-level alignment. Furthermore, as shown in Table 4, hard sample mining (i) significantly boosts the performance of the original algorithm of contrastive instance alignment (e) with a $6.8\%$ mAP. By comparing the models (e-g-i), we can see that the two transformation methods of uniformly dropping and occlusion simulation progressively improve the model performance.

**Comparisons with the self-training pipeline.** We compare our co-training procedure, denoted as CT in Fig. 4, with self-training, denoted as ST, based on the same initialized pretrained source model. In Fig. 4(a1), without updating pseudo-labels, both models fluctuate in the early training stage but 3D-CoCo converges faster and more stably, with higher performance than the self-training model.

Fig. 4(a2) shows the ratio of true positive and false positive predictions, denoted as TPs/FPs. It indicates that our co-training approach derives a lower detection noise than the self-training baseline. Besides, by updating pseudo-labels in the training process, as shown in Fig. 4(b1-b2), the proposed co-training framework consistently outperforms self-training in detection accuracy, and yields an extremely low detection noise with gradually improved pseudo-labels. At last, we conduct sensitivity analyses of pseudo-labels that are filtered by different confidence scores, where lower scores bring in more noisy labels while higher ones tend to miss the positive labels. As shown in Fig. 4(c1), since self-training is fully dependent on pseudo-labels, it is more sensitive to the filtering scores, while our co-training framework is more robust to the quality of pseudo-labels.

## 5 Related Work

**3D point cloud detection.** LiDAR-based 3D detectors aim to localize and classify 3D objects from point clouds, which can be broadly grouped into two categories: point-based and grid-based. The point-based methods [25, 37, 36] take raw points as input and apply PointNet [19] to extract point-wise features and generate proposals for each point. The grid-based methods [32, 33, 14, 34, 43, 38] propose to convert point clouds into regular grids as the model input, in which Voxelization [43, 32] is a common technique to map point clouds into regular 3D voxels. Other methods quantize point clouds into certain kinds of 2D views, such as the bird-eye view [14, 34, 33] and range view [17, 4]. Compared to point-based methods, they are more efficient, accelerating the training on large-scale datasets such as nuScenes [1] and Waymo [26]. In this work, for computational efficiency, we adopt VoxelNet [43] and PointPillars [14] based on the anchor-free detection head [38] as the base detector.

**2D unsupervised domain adaptation.** A variety of solutions have been proposed in 2D vision tasks including classification [5, 10], detection [12, 2] and segmentation [45], which can be roughly divided into two groups: distribution alignment [5, 10, 12] and self-training [45, 9, 11]. For the first group, adversarial learning [15, 7, 12] are leveraged to perform alignment in the feature space. Furthermore, contrastive learning has also been employed for fine-grained feature alignment [10, 31, 24, 40]. Besides, some works borrow image translation techniques to performing alignment at the pixel level [23, 16, 8]. As for the second group, the self-training methods [45, 11, 2, 21] generally assign pseudo-labels to guide the re-training process on the target domain. Compared to these methods, 3D-CoCo originates from the unique properties of the 3D geometry shift. By using the domain-specific encoders, domain-agnostic BEV transformation module, and the transformed point sets, it effectively extends the original contrastive adaptation methods to 3D object detection.

**3D unsupervised domain adaptation.** Some recent works effectively reduce the domain shift in point cloud classification [20] and semantic segmentation [29, 42]. In this paper, we focus on the domain adaptation task of 3D object detection, which has been discussed by only a few works. The statistical normalization method [28] closes the 3D domain shifts by normalizing the object size of the source domain with known statistics of the target domain, which is generally unavailable in unsupervised adaptation tasks. To tackle this issue, SF-UDA [22] uses the temporal coherency to estimate the scales in the target domain, and re-trains the detection model on the target data by transforming the scales of pseudo-labels. Current works [35, 39, 3] further explore the generation mechanisms of pseudo-labels on target data as the supervision signals for self-training. Since the adaptation process of self-training methods can be misguided by noisy pseudo-labels, those methods exploit sophisticated strategies to improve the quality of pseudo-labels. Compared to self-training, 3D-CoCo leverages a co-training framework in which the labeled source data can provide more stable supervisions to the detection model in the adaptation process.

## 6 Conclusion

In this paper, we presented 3D-CoCo for unsupervised domain adaptation of point cloud detection. 3D-CoCo contains a novel model architecture and a new contrastive learning framework. Based on the observation that BEV features are more transferable than low-level geometry features in 3D scenes, we innovatively proposed to integrate domain-specific 3D encoders with a domain-agnostic BEV transformation module. We then conducted contrastive instance alignment, which is augmented by hard sample mining, on the BEV features. The experiments across three autonomous driving datasets showed the effectiveness of 3D-CoCo. As a pilot work for the transfer learning problem of 3D point cloud detection, we follow the typical training setup of unsupervised domain adaptation, which takes more memory footprint than existing self-training methods at training time.

**Acknowledgements.** This work was supported by NSFC (61906119, U19B2035), Shanghai Municipal Science and Technology Major Project (2021SHZDZX0102). Y. Wang was supported by Shanghai Sailing Program and CAAI-Huawei MindSpore Open Fund.

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
