# Learning Transferable Features for Point Cloud Detection via 3D Contrastive Co-training –Supplementary Material–

This supplementary material consists of four parts, including details about the dataset configurations (Sec. 1), technical details of hard sample mining (Sec. 2), implementation details of the proposed 3D-CoCo framework (Sec. 3) and additional experimental results (Sec. 4).

## 1 Datasets

Table 1 provides more details about the datasets used in the paper, including the number of point clouds, sensor configurations, and specific environments, which indicates the existence of the domain shift. Fig. 1 gives three showcases that are randomly selected from the above datasets. It is obvious that the distributions of the 3D patterns are very diverse.

| Dataset | Size | | Sensor | | Environment | | |
|---|---|---|---|---|---|---|---|
| | #Training | #Validation | LiDAR Type | Beam Angles | Location | Rainy | Night |
| Waymo | 158081 | 39987 | 1×64+4×200-beam | [-24°, 4°] | USA | Yes | Yes |
| nuScenes | 28130 | 6018 | 1×32-beam | [-16°,11°] | USA, Singapore | Yes | Yes |
| KITTI | 3712 | 3769 | 1×64-beam | [-24°,4°] | Germany | No | No |

Table 1: Details of the datasets that indicate the existence of the domain shift.

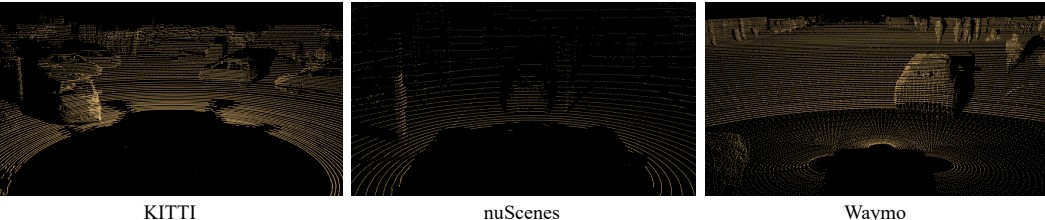

| KITTI | nuScenes | Waymo |

Figure 1: The visualization of LiDAR point clouds from the frontal view.

## 2 Hard Sample Mining

To construct hard samples for contrastive alignment, we transform the point clouds of the target domain from two aspects, *i.e.*, object density and object completeness.

**Object density.** For each point $p_n = (x_n, y_n, z_n, i_n)$ of a point cloud $P = (p_1, p_2, \ldots, p_N)$, we first calculate its elevation angle $\theta_n$ and the perspective angle $\phi_n$ to the LiDAR sensor:

$$\theta_n = \arccos\left(\frac{\sqrt{x_n^2 + y_n^2}}{\sqrt{x_n^2 + y_n^2 + z_n^2}}\right), \quad \phi_n = \arctan(\frac{y_n}{x_n}). \tag{1}$$

We then arrange the points according to their elevation angles $\theta$ and use a sampling interval $\delta$ to slice them into $L$ separate lines as virtual laser beams, where $\delta$ is set to $0.4$ for KITTI and $1.3$ for nuScenes. For Waymo data where point clouds are transferred from range images, we directly extract virtual laser beams according to the corresponding horizontal axis of range coordinate. At last, we uniformly sample the virtual beams by random step $\Delta_L$ to reserve $L'$ beams, where

$$L' = \lceil L/\Delta_L \rceil, \ \Delta_L \in [1, L). \tag{2}$$

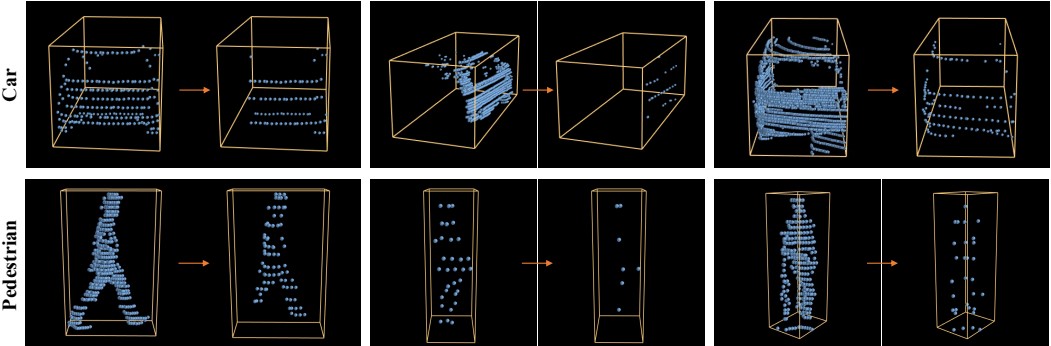

Figure 2: The visualization of mined hard examples for the categories of Car and Pedestrian.

Specifically, considering that Waymo integrates points from 5 sensors, we process data of each sensor respectively and integrate the transformed points as results.

**Object completeness.** We denote the perspective range of $P$ as $\Delta\phi$. To simulate the patterns of severe occlusions, we transform $P$ by randomly removing the points that locate within a subinterval of $\Delta\phi$, where

$$
\begin{aligned}
\Delta\phi &= \phi_{\max} - \phi_{\min} \\
\phi_{\min} &= \min(\{\phi_n\}_{n=1:N}) \\
\phi_{\max} &= \max(\{\phi_n\}_{n=1:N}).
\end{aligned}
\tag{3}
$$

Notably, in real scenes, hard samples tend to appear at a great distance, therefore we also move the transformed patterns to a remote location $c'$ according to

$$
c' = (\Delta_L \times c_x, \Delta_L \times c_y, c_z),
\tag{4}
$$

where $c = (c_x, c_y, c_z)$ is the original center of the point cloud $P$. Fig. 2 provides more showcases of the transformed point clouds.

## 3 Implementation Details

We use the complete training and validation sets of nuScenes and KITTI, and sample $1/5$ training scenes and $1/4$ validation scenes for Waymo. For all datasets, the coordinate origins are shifted to the ground plane and the detection range is set to $[-2m, 4m]$ for the $Z$ axis. For the other two axes, the detection ranges are $([0m, 70.4m], [-40m, 40m])$ for KITTI, $([-51.2m, 51.2m], [-51.2m, 51.2m])$ for nuScenes, and $([-75.2m, 75.2m], [-75.2m, 75.2m])$ for Waymo.

At training time, we augment the datasets by applying point cloud flipping (along the $X$ and $Y$ axes), global scaling, global rotation, and random global translation (the entire point cloud scene is moved by a random distance) to the raw point clouds. We also adopt the GT-Sampling data augmentation strategy from [2], which pastes the ground-truth boxes and their inside points from other scenes to the same locations of current training scenes.

As mentioned in the main manuscript, the progressive training procedure is the key to improve the quality of pseudo-labels. Specifically, after the warm-up process, we update pseudo-labels every 3 epochs for KITTI. In order to save the computation time of generating pseudo-labels on large datasets, we update pseudo-labels every 5 epochs for nuScenes and Waymo.

## 4 Additional Experimental Results

**t-SNE visualization.** In Fig. 3, we use t-SNE [1] to visualize the distribution of features from the source and target domains, produced by three models (*i.e.*, the baseline model trained on the source domain, the self-training method, and 3D-CoCo). Due to the domain gap, as shown by the Source Only model, the features of different categories (*i.e.*, background or foreground car) in two domains

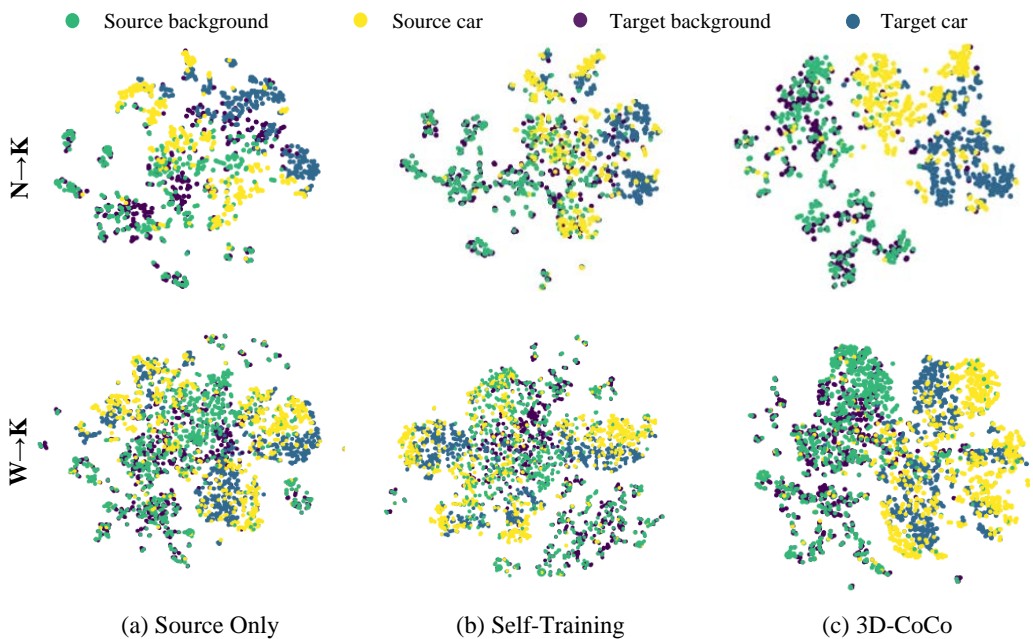

(a) Source Only            (b) Self-Training           (c) 3D-CoCo

Figure 3: The t-SNE visualization of sample features. **(a)** The source only model is trained without any adaptation technique. **(b)** The self-training method applies the pseudo-labels on target data for re-training, but it fails to align the distribution of features from the source and target domains without any access to the source data. **(c)** The proposed 3D-CoCo framework leverages both labeled source data and unlabeled target data for co-training, which obviously enhances the intra-class compactness and encourages inter-class separability in the feature space. **N:** nuScenes; **K:** KITTI; **W:** Waymo.

| Task | Method | VoxelNet | | | |
| | | $AP_{BEV}$ | Closed Gap | $AP_{3D}$ | Closed Gap |
|---|---|---|---|---|---|
| | Source Only | 17.1 | - | 12.7 | - |
| N → K | 3D-CoCo | 27.2 | +38.40% | 24.9 | +46.92% |
| | Oracle | 43.4 | - | 38.7 | - |
| | Source Only | 48.3 | - | 45.3 | - |
| W → K | 3D-CoCo | 38.1 | - | 36.0 | - |
| | Oracle | 43.4 | - | 38.7 | - |
| | Source Only | 16.0 | - | 13.6 | - |
| W → N | 3D-CoCo | 19.5 | +18.23% | 15.9 | +16.91% |
| | Oracle | 35.2 | - | 27.2 | - |
| | Source Only | 9.1 | - | 9.1 | - |
| N → W | 3D-CoCo | 21.8 | +21.38% | 15.9 | +12.93% |
| | Oracle | 68.5 | - | 61.7 | - |

Table 2: Adaptation results on the Pedestrian category. **N:** nuScenes; **K:** KITTI; **W:** Waymo.

are heavily overlapped, which indicates poor generalization performance. After domain adaptation, we observe that the feature distribution of the self-training method is still irregular, while that of 3D-CoCo shows better clustering properties, in the sense that the features of the same category in the two domains are better aligned and those from different categories are separated more clearly.

**The Pedestrian category.** In addition to the *Car* category that is shown in the main manuscript, we here provide more experimental results on the *Pedestrian* category. Table 2 gives the adaptation results of 3D-CoCo with the VoxelNet encoder. It validates the effectiveness of 3D-CoCo with consistent improvements over the Source Only model on a variety of adaptation tasks, including nuScenes→KITTI, nuScenes→Waymo, and Waymo→nuScenes. The only exception on Waymo→KITTI, where the Source Only model outperforms 3D-CoCo and even the Oracle model, is largely caused by the limited number of training samples of Pedestrian on the target KITTI dataset.

| Methods | $AP_{BEV}$ | $AP_{3D}$ |
|---|---|---|
| Pooling-based | 75.8 | 62.9 |
| Keypoint-based (ours) | 77.1 | 65.6 |

Table 3: Comparison of extraction methods.

| Methods | $AP_{BEV}$ | $AP_{3D}$ |
|---|---|---|
| $R = 3$ | 78.4 | 64.1 |
| $R = 5$ | 75.9 | 62.4 |
| $R = 7$ | 77.1 | 65.6 |

Table 4: Sensitivity analysis of $R$.

| Methods | $AP_{BEV}$ | $AP_{3D}$ |
|---|---|---|
| $\tau = 0.01$ | 75.7 | 61.7 |
| $\tau = 0.07$ | 77.1 | 65.6 |
| $\tau = 0.2$ | 74.7 | 61.4 |

Table 5: Sensitivity analysis of $\tau$.

| Methods | $AP_{BEV}$ | $AP_{3D}$ |
|---|---|---|
| $\lambda = 0.25$ | 76.0 | 62.6 |
| $\lambda = 0.5$ | 77.1 | 65.6 |
| $\lambda = 1.0$ | 76.1 | 63.5 |

Table 6: Sensitivity analysis of $\lambda$.

**Comparison of extraction methods.** We use another feature extraction method based on average pooling as a compared method on the nuScenes→KITTI benchmark in Table 3. We believe that the average pooling method tends to result in more ambiguous features.

**Sensitivity analysis.** We conduct extra sensitivity analysis on other hyper-parameters, including the sample number of keypoints $R$, temperature parameter $\tau$ and the loss weight $\lambda$ of adaptation loss. We show the results in Table 4, 5 and 6.

**Analysis of error bars.** We run 3D-CoCo and the self-training method for 5 times. As shown in Fig. 4, compared to the self-training method, 3D-CoCo achieves a higher median value of the accuracy and a more concentrated distribution of the results. It indicates that 3D-CoCo not only boosts accuracy but also performs more stably.

**Qualitative results.** We show the qualitative results in Fig. 5, which illustrates that 3D-CoCo improves the adaptive detection performance by greatly reducing the false positive predictions and increasing the localization accuracy. Specifically, when adapted from sparse source domain (*i.e.*, nuScenes) to dense target domain (*i.e.*, Waymo or KITTI), the Source Only model tends to easily produce false positives due to the increase of point cloud density, while 3D-CoCo can effectively avoid those erroneous predictions. Besides, due to changes in the physical environments and the size of objects, the domain shifts are also reflected in the inaccurate 3D bounding boxes produced by the Source Only model. We observe that our proposed 3D-CoCo achieves higher localization accuracy of the 3D bounding boxes.

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

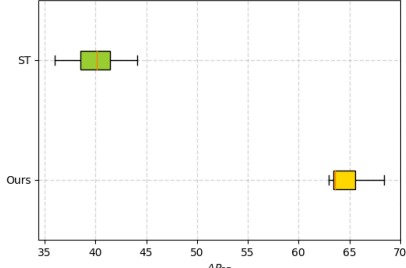
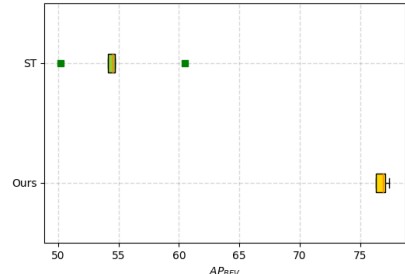

Figure 4: The box-plot on $AP_{3D}$ (**Left**) and $AP_{BEV}$ (**Right**) for 3D-CoCo and self-training (ST).

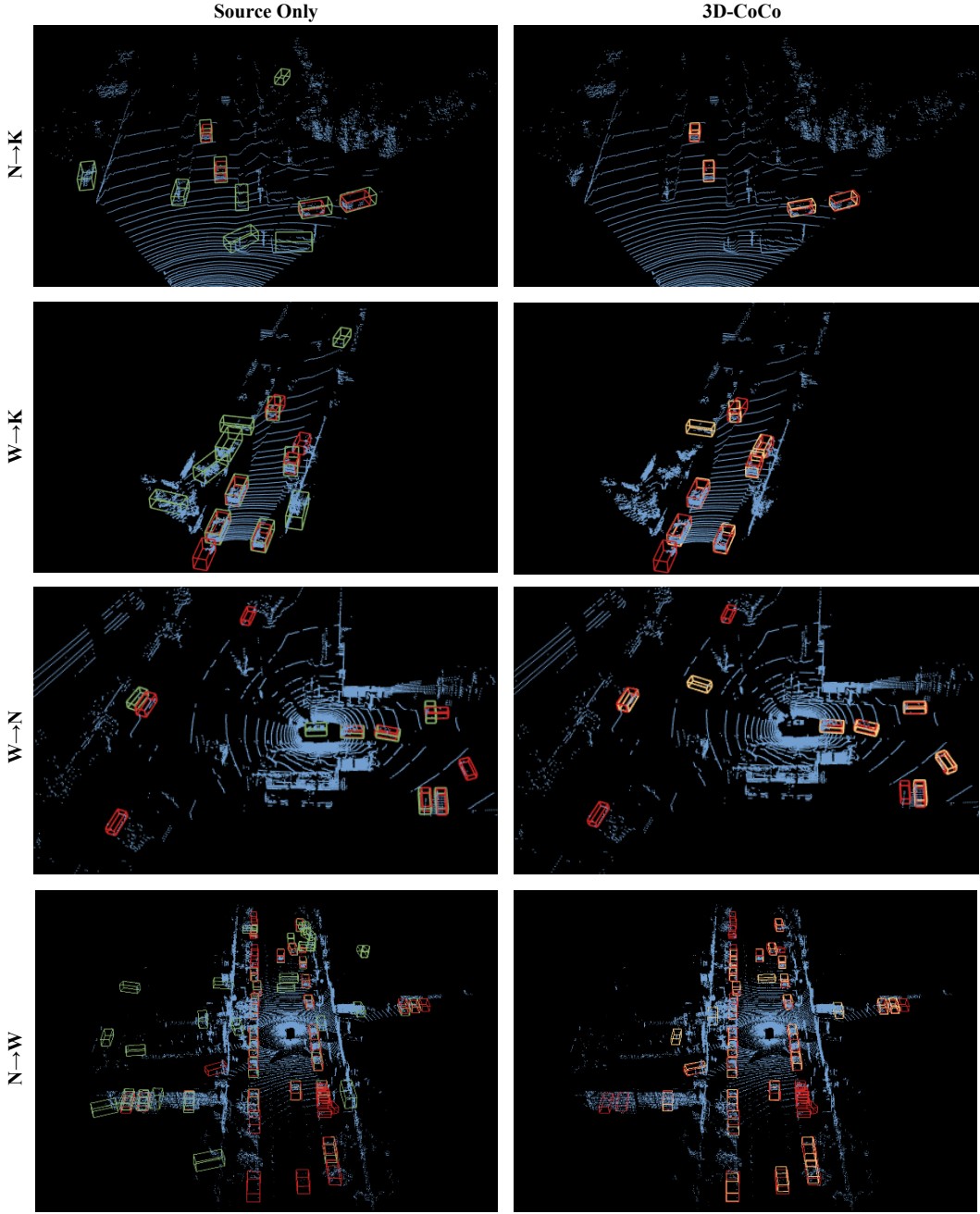

Figure 5: Qualitative results on four adaptation tasks. **N:** nuScenes; **K:** KITTI; **W:** Waymo. **Red:** Ground-truth; **Green:** Predictions by the Source Only model; **Yellow:** Predictions by 3D-CoCo. Obviously, the predictions of 3D-CoCo align better with the ground-truth bounding boxes.