# OpenReview forum: "Learning Transferable Features for Point Cloud Detection via 3D Contrastive Co-training"
_NeurIPS.cc/2021/Conference — NeurIPS 2021 Poster_

### Official Review · Reviewer_2DjX · 2021-07-16

**Rating:** 6
**Confidence:** 4

**Summary:**

This paper contributes as an early study of unsupervised domain adaptation (UDA) on 3D point cloud detection, an important step towards zero cost of labeling. Different from 2D image data that have regular grid structures, 3D point cloud data could have huge geometric variations such as point density and occlusion due to different physical environments or LiDAR sensor configurations; consequently, the low-level features of point cloud representation may be not transferable across domains. Taking it into account, this paper uses two individual encoders for the source and target domains to capture domain-specific characteristics and a shared Bird-Eye-View (BEV) transformation module to learn domain-agnostic BEV features, which have similar grid structures as images and thus are more transferable. Specifically, this paper extends contrastive instance alignment to point cloud detection, based on BEW features of foreground and background proposals. Direct use of contrastive instance alignment can cause the mismatch in point density and occlusion ratio between pseudo-labeled and true samples distributions in the target domain. To address it, this paper proposes to augment positive easy target samples (with dense points and complete geometry) by their artificial counterparts with sparse points and occlusion. Moreover, this paper constructs a new benchmark using three large-scale datasets. Empirical results show that the proposed method is effective and achieves the new state of the art.

**Ethical Concerns:**

There are no ethical issues with this paper.

**Limitations And Societal Impact:**

Annotating point clouds is more difficult and expensive than annotating images for object detection tasks. Good 3D UDA methods are desired to reduce the annotation efforts. On the constructed new benchmark of three autonomous driving datasets, this paper achieves a significant gain in detection performance, one step closer to the goal. As for the limitation, this paper has mentioned that the proposed method takes more memory footprint than existing self-training methods at the training time whereas does not address it. The potential negative societal impact of their work has not been mentioned. The suggestions for improvement are provided in Main Review.

**Main Review:**

Although the contributions and novelty of this paper are considerable, there are still many problems.

1. The full name should be mentioned when BEV is shown for the first time in Section 1, where it would be better to provide a brief description of BEV and its related references for a wider range of audiences.

2. It would be better to compare the high-level 3D features with the BEV features. The high-level 3D features are extracted by the same but deeper network layers than the low-level ones. The 3D domain shift is mapped into the 2D one, causing information loss.  Would it reduce the feature discriminability? Would it affect the single-domain detection performance, e.g., cases of a general detection setup in individual domains? It is also interesting to know how the features extracted by spherical convolutions [a] behave.

3. The idea of separate encoders is similar to that of [b,c,d] for 2D UDA. The authors should consider them as related works and make comparisons and discussions appropriately.

4.  It is interesting to approximately quantify the distribution discrepancy across domains based on the performances of passing source samples through the target encoder and passing target samples through the source encoder. In Section 4.3, the in-domain performance is evaluated on the target domain; however, according to the previous descriptions, the cross-domain performance should be evaluated on the target domain. There exists a contradiction and a clarification is required.

5. Does the "co-training" in this paper mean two individual encoders for the source and target domains? Or the training strategy? This paper also adopts the pseudo-labeling technique from self-training. In Lines 56-59, the difference is not clear enough.

6. In Lines 109-112, "bi-directional knowledge sharing" is similar to multi-task learning [e]? Why does it hold? Are there any related works that support this argument? In the ablation studies, it seems that the random object scaling has a great effect on the source performance and this has nothing to do with the proposed method.

7. In Algorithm 1, this paper conducts three-stage training. why not two stages? The warm-up stage introduces another set of training hyperparameters. Besides, the augmented target set is denoted by the original one? The notations should be different.

8. It would be more clear to expand the losses of L_cls^S, L_loc^S, and L_reg^T.

9. In Section 3.2, this paper uses a keypoint-based feature extraction approach. Comparisons with other kinds of feature extraction approaches are needed. Besides, the model sensitivity to the hyperparameter R is to be examined.

10. In Eq. (5), some notations are confusing, such as the summation range in the denominator in L_intra and L_inter. A negative sign is missing in L_intra and L_inter? Otherwise, samples of the same classes are pushed away? In L_intra, samples of other classes should be used as negative ones whereas this paper does not make it clear. In L_inter, other samples in the same class are used as negative ones? Is this the typical contrastive learning method of instance discrimination from self-supervised learning? How can this method increase the inter-class distance between different categories within the same domain? Are there any references for contrastive instance alignment? The main component of the proposed algorithm is so ambiguous that the experimental results may be not so convincing.

11. In Eq. (5), the domain \mathcal{D} is similar to model D; it would be better to use different notations. In Eq. (6), "arg" is redundant. The model sensitivity to the temperature \tao and the hyperparameter \lambda is to be examined.

12. In Line 159, is there a specific rule to distinguish between easy and hard samples? Please make it clear. In Line 166, how many points are discarded? In Line 169, how to calculate the viewpoint of a certain sample? How many viewpoints do you select to discard? Are there any solid principles to make decisions? Are there any related references? Please make it clear.

13. In Lines 170-173, the proposed augmentation can also be done in the source domain since source samples can be viewed as easy ones. Consequently, the distribution mismatch between the source and target domains can be further reduced. It is interesting to see whether the model performance can be further improved. The proposed augmentation strategy is somewhat similar to consistency regularization [f,g] from semi-supervised learning, which enforces consistent predictions between weakly and strongly augmented versions of the same sample image. Maybe the related works should be considered and discussed.

14. In Line 179, can you describe how to update the pseudo labels in detail?

15. Is each domain split into training and test sets? Is the performance evaluated on the training or test set of the target domain?


[a] H. Lei, N. Akhtar and A. Mian, "Spherical Kernel for Efficient Graph Convolution on 3D Point Clouds," in IEEE Transactions on Pattern Analysis and Machine Intelligence, doi: 10.1109/TPAMI.2020.2983410.

[b] Konstantinos Bousmalis, George Trigeorgis, Nathan Silberman, Dilip Krishnan, and Dumitru Erhan. 2016. Domain separation networks. In Proceedings of the 30th International Conference on Neural Information Processing Systems (NIPS'16). Curran Associates Inc., Red Hook, NY, USA, 343–351.

[c] E. Tzeng, J. Hoffman, K. Saenko and T. Darrell, "Adversarial Discriminative Domain Adaptation," 2017 IEEE Conference on Computer Vision and Pattern Recognition (CVPR), 2017, pp. 2962-2971, doi: 10.1109/CVPR.2017.316.

[d] W. Chang, H. Wang, W. Peng and W. Chiu, "All About Structure: Adapting Structural Information Across Domains for Boosting Semantic Segmentation," 2019 IEEE/CVF Conference on Computer Vision and Pattern Recognition (CVPR), 2019, pp. 1900-1909, doi: 10.1109/CVPR.2019.00200.

[e] Caruana, R. Multitask Learning. Machine Learning 28, 41–75 (1997). https://doi.org/10.1023/A:1007379606734.

[f] D. Berthelot, N. Carlini, E. D. Cubuk, A. Kurakin, K. Sohn, H. Zhang, and C. Raffel. Remixmatch: Semi-supervised learning with distribution matching and augmentation anchoring. In Proc. Int. Conf. on Learn. Rep., 2020.

[g] S. Roy, A. Siarohin, E. Sangineto, S. R. Bulò, N. Sebe and E. Ricci, "Unsupervised Domain Adaptation Using Feature-Whitening and Consensus Loss," 2019 IEEE/CVF Conference on Computer Vision and Pattern Recognition (CVPR), 2019, pp. 9463-9472, doi: 10.1109/CVPR.2019.00970.

**Time Spent Reviewing:**

5

---

> ### Author Response · Authors · 2021-08-10
> **Our Response to Reviewer 2DjX**
>
> We thank the reviewer for the constructive comments.
>
> Q1. Description of BEV.
>
> BEV is short for Bird's Eye View. The BEV feature is widely used as a form of representation in 3D detection [14,32,33,34,43], especially for autonomous driving. It represents the scene from a view of a high angle that has the potential to avoid scale ambiguity and occlusions of the objects.
>
> &nbsp;
>
> Q2. (1) 3D features vs. BEV features; (2) Spherical convolutions.
>
> (1) It is a common practice in existing 3D detection methods [14,32,34,43] to convert 3D LiDAR points to BEV features, which have the following properties:
>
> - High efficiency: The use of BEV features greatly accelerates the training process, especially compared with point-based methods that generate proposals from each LiDAR point. In practice, we find that training a detection model based on 3D features is very time-consuming and not affordable for large-scale datasets such as nuScenes [1] and Waymo [26].
>
> - Negligible information loss in cross-domain scenarios: In Fig. 1, the 3D domain shift tackled by this paper is mainly caused by different sensor configurations or environments, instead of the intrinsic geometry of objects. Although the transformation of 3D features into 2D space inevitably leads to the information loss in the height dimension, in real auto-driving scenarios, such a loss would not enlarge the domain shift. It is negligible as the domain shift does not mainly exist in the height dimension of the object geometry, which can be learned and inferred from BEV features.
>
> - Single-domain discriminability: The BEV features can offer more compact and global representations of the scene, making it easier for the model to deal with object occlusion. Therefore, it does not reduce the discriminability of the features, nor single-domain performance.
>
> (2) As a classic technique for learning 3D features, the spherical convolution is based on a spherical kernel that updates the features of each point through a graph network. Although it is effective for learning 3D geometry, we find it too computation-intensive to process datasets used in this paper that have larger amounts of LiDAR points.
>
> &nbsp;
>
> Q3. The idea of separate encoders.
>
> Some existing methods for image domain adaptation also use separate encoders to learn domain-specific features [b,c,d], which will be included as related work in the revision. However, to the best of our knowledge, 3D-CoCo is the first one to use such an architecture in 3D scenarios. Different from the previous literature in this field [35], here, the use of separate encoders is motivated by our key observation that  3D transfer learning is hampered by the lack of transferability in low-level 3D features. Further, the separate 3D encoders serve as the base of the entire co-training framework. From an empirical view, they enable 3D-CoCo to improve both in-domain discriminability and cross-domain transferability of the features.
>
> &nbsp;
>
> Q4. In-domain and cross-domain performance.
>
> In Section 4.3, after the co-training phase, we evaluate the model, which has separate encoders and a domain-agnostic BEV transformation module, respectively on the source domain test set (in-domain) and the target domain test set (cross-domain).
>
> In Table 2, we show that the co-training improves the performance on both test sets, which may be counter-intuitive as most domain adaptation methods compromise between the in-domain discriminability and cross-domain transferability of the learned features.
>
> &nbsp;
>
> Q5. (1) Definition of co-training; (2) Co-training vs. self-training.
>
> (1) Co-training refers to the entire training strategy in Alg. 1 that jointly uses labeled source data and unlabeled target data during the training phase. This is done by matching similar source and target samples and closing their feature distance through the contrastive loss.
>
> (2) Please refer to our response to Reviewer qi8q (Q2).
>
> &nbsp;
>
> Q6. (1) Bi-directional knowledge sharing; (2) Random object scaling (ROS).
>
> (1) Empirically, bi-directional knowledge sharing (Line 112) means that the co-training improves the performance on both source and target test sets. Based on the separate encoders, this is mainly realized by performing contrastive alignment between source and target data, and optimizing the model with data input from different domains under a unified contrast loss term. Thus, the so-called bi-directional knowledge sharing is different from typical multi-task learning methods.
>
> (2) From SN [28], the different object size is an essential part of domain shift. To this end, we use the ROS from ST3D [35] as an alternative technique upon 3D-CoCo, which leads to a tradeoff between in-domain and cross-domain performance. In Table 2, for a comprehensive comparison, we use two different ranges of the ROS factor, one for better adaptation results, while the other for better source performance. We observe that in both settings, 3D-CoCo outperforms the Source Only baseline remarkably on the target test set.
>
> &nbsp;
>
> Q7. (1) The warm-up stage; (2) Notation of augmented target set.
>
> (1) In Fig. 4 (a1), the performance fluctuates in the first 15 epochs and then converges stably before updating the pseudo-labels. Thus, warming-up the model is necessary. It can improve the efficacy of the first pseudo-labels update stage by avoiding a very ill-defined detection model. An empirical parameter for ending the warm-up stage is at half of the total epochs.
>
> (2) Thanks for the suggestion. We'll fix it in the revision.
>
> &nbsp;
>
> Q8. Expanding $L_\text{cls}^S$, $L_\text{loc}^S$, and $L_\text{reg}^T$.
>
> **Please use Safari or Firefox to display the equations in LaTeX properly.**
>
> We use CenterPoint [38] as the base detector to predict Gaussian heatmaps at each annotated object center $q_i$ for each class $c_i\in \{1\cdot\cdot\cdot |C|\}$, where $i$ is the object index. For pixel position $p$ in the heatmap of category $k$, we have $Y_{p,k} = \max_{i:c_i=k}\exp(-\frac{{(p-q_i)}^2}{2\sigma_i^2})$, where $\sigma_i$ is the size of the object, and $\max_{i:c_i=k}$ covers all objects under category $k$. We define $L_\text{cls}^S$ in a form of focal loss:
>
> $$L_\text{cls}^S =  -\frac{1}{N^S}\sum_{p,k} (1-\hat Y_{p,k}^S)^{\alpha} \log(\hat Y_{p,k}^S) , \text{if~} Y_{p,k}^S = 1,$$
>
> $$L_\text{cls}^S = -\frac{1}{N^S} \sum_{p,k} (1-Y_{p,k}^S)^{\beta}(\hat Y_{p,k}^S)^{\alpha}\log(1-\hat Y_{p,k}^S), \text{otherwise},$$
>
> where $\alpha=2$, $\beta=4$, and $N^S$ is the number of objects.
>
> The other two loss terms regress the size map $Z$, center offset map $O$, and rotation map $A$:
>
> $$L_\text{loc}^S = \frac{1}{N^S}\sum_{i=1}^{N^S}\lambda_X|\widehat{X}_{q_i}^S -X_i^S|, X = \{Z,O,A\}.$$
>
> $$L_\text{reg}^T = \frac{1}{N^T}\sum_{i=1}^{N^T}\lambda_X|\widehat{X}_{q_i}^T -X_i^T|, X = \{Z,O,A\}.$$
>
> &nbsp;
>
> Q9. (1) Feature extraction methods; (2) Analyses of $R$.
>
> (1) We use another feature extraction method based on average pooling as a compared method.
>
> On the nuScenes $\rightarrow$ KITTI benchmark:
>
> | Method | $\text{AP}_\text{BEV}$ (%) | $\text{AP}_\text{3D}$ (%)
> | :-----| ----: | :----:
> | Pooling-based | 75.8 | 62.9
> | Keypoint-based (ours) | 77.1 | 65.6
>
> (2) The sensitivity analysis of $R$:
>
> | Method | $\text{AP}_\text{BEV}$ | $\text{AP}_\text{3D}$
> | :-----| ----: | :----:
> | R=3 | 78.4 | 64.1
> | R=5 | 75.9 | 62.4
> | R=7 | 77.1 | 65.6
> | Source only | 45.2 | 32.6
>
> &nbsp;
>
> Q10. Notations in Eq. (5).
>
> In our response to Reviewer qi8q (Q1), we correct the notations of Eq. (5) and provide detailed explanations about the motivation of the contrastive loss. A similar idea of contrastive instance alignment has also been applied in 2D tasks [31].
>
> &nbsp;
>
> Q11. (1) Notation; (2) Sensitivity analyses of $\tau$ and $\lambda$.
>
> (1) Thanks for the suggestion. We'll fix it in the revision.
>
> (2) On nuScenes $\rightarrow$ KITTI, by setting $\lambda$ to 0.5:
>
> | Method | $\text{AP}_\text{BEV}$ | $\text{AP}_\text{3D}$ |
> | :-----| ----: | :----:
> | $\tau$ = 0.01 | 75.7 | 61.7
> | $\tau$ = 0.07 | 77.1 | 65.6
> | $\tau$ = 0.2 | 74.7 | 61.4
> |Source only | 45.2 | 32.6
>
> By setting $\tau$ = 0.07:
>
> | Method | $\text{AP}_\text{BEV}$ | $\text{AP}_\text{3D}$ |
> | :-----| ----: | :----:
> | $\lambda$ = 0.25|76.0|62.6
> | $\lambda$ = 0.5|77.1|65.6
> | $\lambda$ = 1.0|76.1|63.5
>
> &nbsp;
>
> Q12. Hard sample mining (HSM).
>
> Line 159: We do not use any specific metric to quantify and differentiate easy/hard samples, but intuitively consider the specific forms of possible geometric domain shift for HSM,.
>
> Lines 166/169: We describe how to drop out points and how to select viewpoints for HSM in the supplementary material.
>
> Other related references include point cloud data augmentation methods [32, A], which enhance the data diversity to improve the single-domain performance, while we use the transformed hard samples to further reduce the domain shift.
>
> &nbsp;
>
> Q13. (1) HSM on source domain; (2) HSM vs. the consistency regularization [f,g].
>
> (1) We agree that augmenting the source domain is an effective way to improve generalization ability. However, the proposed HSM specifically focuses on reducing the distribution mismatch caused by the limited diversity of target pseudo-labels. HSM can be combined with any other augmentation methods performed on the source domain.
>
> (2) The consistency regularization introduces data perturbations and forces the model output to remain unchanged. These methods use common augmentation strategies (e.g., flipping and cropping). By contrast, our method creates fictitious hard samples by considering the priors of geometry variations.
>
> &nbsp;
>
> Q14. Pseudo-label update.
>
> Please refer to our response to reviewer RhUF (Q1).
>
> &nbsp;
>
> Q15. Dataset splits.
>
> In accordance with the common practice [35], each domain is split into training and validation sets, and we evaluate 3D-CoCo on the validation set of target domains.
>
> &nbsp;
>
> [A] Choi, J.,  Y. Song, and  N. Kwak. "Part-Aware Data Augmentation for 3D Object Detection in Point Cloud." (2020).

---

> > ### Comment · Reviewer_2DjX · 2021-08-18
> > **Thanks for the response!**
> >
> > Thanks for the response! It solves most of my problems. There are a few minor comments:
> > 1. Are cosine distance and cosine similarity the same concepts? Please improve the accuracy of the article wording.
> > 2. A negative sign should be added to each term in Eq. (5). Please refer to [a].
> > 3. The proposed co-training strategy includes the self-training technique (i.e., pseudo label generation). The use of self-training does not affect the significance of the main contributions. There is no need to spend a lot of space describing the differences between them.
> > 4. Please carefully proofread the paper to keep the coherence of the context.
> >
> >
> > [a] Ting Chen, Simon Kornblith, Mohammad Norouzi, and Geoffrey Hinton. "A Simple Framework for Contrastive Learning of Visual Representations." (2020)

---

> > > ### Author Response · Authors · 2021-08-20
> > > **Thanks for the comments!**
> > >
> > > Dear reviewer, thanks again for your comments.
> > >
> > > 1. For a clarification, in Eq.  (4), we search positive sample pairs by minimizing the cosine distance between the features of $I_i^S$ and $I_{j}^T$. Therefore, in Line 144, the cosine distance should be defined as $\Phi(\mathbf{x},\mathbf{y}) = 1 - \text{CosineSimilarity}(\mathbf{x},\mathbf{y})$ instead of $\Phi(\mathbf{x},\mathbf{y}) = \text{CosineSimilarity}(\mathbf{x},\mathbf{y})$. Thank you so much for bringing this to our attention.
> > >
> > >
> > > 2. Yes, we agree with that, and we will fix it in the revision.
> > >
> > >
> > > 3. Thanks for the suggestion. Indeed, co-training follows the technical details of self-training to generate pseudo-labels. It improves the latter through effective utilization of labeled source data and the hard sample mining method. We will make the comparison more concise and clear in the revision, leaving the space for adding the content included in the rebuttal.
> > >
> > >
> > > 4. Admittedly, the current version of the manuscript has a few typoes. We thank all reviewers for pointing out these problems, and we will proofread this paper and fix them carefully.

---

> > > > ### Comment · Reviewer_2DjX · 2021-08-27
> > > > **Thanks for the response!**
> > > >
> > > > Thanks for the response! All my concerns have been addressed.

---

### Official Review · Reviewer_RhUF · 2021-07-17

**Rating:** 6
**Confidence:** 4

**Summary:**

The paper proposed an unsupervised domain adaptation method for point cloud detection. The method is built on several effective strategies, including BEV feature representation, pseudo labels, instance contrastive alignment and hard mining.

**Limitations And Societal Impact:**

1) Details about the pseudo label generation need to be described.  How did you run Eq.(4)?

2) How to process or avoid the possible errors of the pseudo labels? How do such pseudo-label errors influence the final results? It’s good to have some analysis about the relationship between the quality of the pseudo labels and the final performance of the proposed model.

3) Synthetic/simulation-to-real generalization experiments are interesting. If possible, it would be nice to see some tests under this setting.


**Main Review:**

Novelty: the instance contrastive joint training is interesting and novel according to my knowledge.

Significance: It clearly improves the baseline (trained only on source domain) and outperforms the existing unsupervised domain adaptation methods.

Post Rebuttal: My concerns about pseudo labels have been addressed in the rebuttal. My final opinion: this is a valuable paper but not a breakthrough in the field.

**Time Spent Reviewing:**

3

---

> ### Author Response · Authors · 2021-08-10
> **Our Response to Reviewer RhUF**
>
> We thank the reviewer for the constructive comments.
>
> Q1. (1) Details about pseudo-label generation; (2) How to run Eq. (4)?
>
> (1) We follow the pseudo-label generation process of ST3D [35], which has two parts. First, before the warm-up training stage, we use the source detection model to initialize the pseudo-labels. Second, after the warm-up stage, we progressively refine the pseudo-labels at each training epoch, which specifically has two steps:
>
> - Matching and merging the currently updated pseudo-labels with the previous ones (termed Memory Ensemble by ST3D): This is done by calculating a 3D IoU matrix of the object detection boxes under two adjacent updates of pseudo-labels, and treating pairs that exceed a certain threshold as matching pairs. We then take the boxes with higher confidence as the current pseudo-labels.
>
> - Marking the status of the unmatched pseudo boxes of the first step to determine whether to discard them (termed Memory Voting by ST3D): When a new unmatched box appears, we assume that it is a correct sample caused by the model update, and we thus save the box and record its the number of its unsuccessful matches. If the box is unmatched for a long time after several epochs, it is considered to be an incorrect sample and will be permanently deleted.
>
> (2) Given a source sample $I_i^S$, Eq. (4) searches for its positive target sample $I_{j^\star}^T$ to optimize the contrastive loss in Eq. (5). Concretely, we calculate the cosine distances, denoted by $\Phi(\cdot, \cdot)$, between $I_i^S$ and all target samples under the same category $c$ (indicated by the same pseudo-labels), among which we select the one with the smallest distance as $I_{j^\star}^T$, where $j^\star = \mathop{\arg\min}\limits_{1\leq j\leq N_c^T}\{\Phi(I_i^S, I_j^T)\}$. Please use the Safari or Firefox browser to display the equations in LaTeX properly.
>
> &nbsp;
>
> Q2. (1) How to process or avoid the possible errors of pseudo-labels? (2) Analyses of the relationship between the quality of pseudo-labels and the final performance.
>
> (1) In our domain adaptation scenarios, the errors of pseudo-labels are mainly caused by the distribution shift across domains. In 3D-CoCo, they are processed in three ways:
>
> - The progressive updates of pseudo-labels are helpful to improve the quality of pseudo-labels and eliminate the existing errors (as explained in the above response).
>
> - The proposed co-training framework uses the source labeled data as strong guidance, which has great effects on the feature learning process of the target data. Therefore, the quality of pseudo-labels is gradually improved during the training phase.
>
> - The proposed method of hard sample mining is helpful to enrich the pseudo-labels and can further avoid errors of pseudo-labels caused by the distribution shift across domains.
>
> (2) The visualization in Fig. 4 shows that the three contributions mentioned above are indeed helpful to avoid the errors of pseudo-labels, and the final performance is largely dependent on the quality of pseudo-labels. More specifically,
>
> - We use the ratio of true positive and false positive predictions (#TPs/#FPs) to indicate the quality of pseudo-labels. By comparing the final performance in Fig. 4 (a1) and (b1), as well as the corresponding quality of pseudo-labels in (a2) and (b2), we can see that high-quality pseudo-labels lead to better detection results.
>
> - The progressive refinement of pseudo-labels is helpful to improve the label quality and the detection results. The strong evidence is that at around the 15th training epoch in Fig. 4 (b2) after the warm-up stage, the green curve (i.e., co-training with label update) and the blue curve (i.e., co-training with label update and hard sample mining) grow rapidly, and the model performance in (b1) grows simultaneously. During these periods, the pseudo-labels are progressively refined with hard sample mining in each training epoch. Please refer to our response to Reviewer DPJL for more analyses.
>
> &nbsp;
>
> Q3. Simulation to real.
>
> We agree that the simulation-to-real is an interesting transfer learning setup. However, due to time constraints, we plan to conduct such experiments in future work and hope to provide results in the revision.

---

### Official Review · Reviewer_DPJL · 2021-07-18

**Rating:** 6
**Confidence:** 4

**Summary:**

This paper proposes a domain adaptation point cloud object detection method termed 3D-COCO to deal with geometry shifts caused by different physical environments or LiDAR sensors. The authors use the shared BEV module to learn the domain-invariant for both source and target domain, while the augmented contrastive alignment schemes are also introduced to discover the hard samples. The mismatch is greatly reduced by 3D-CoCo with transformed point clouds. The experimental results on three benchmarks demonstrate the effectiveness of the proposed method.

**Limitations And Societal Impact:**

The authors have pointed out the limitations of their work.

Since it is just a basic method to deal with 3D point cloud detection, it has nothing to do with the potential negative societal impact.

**Main Review:**

Strength：

+ The paper is well written and easy to follow.
+ The proposed framework seems simple yet effective.
+ The thinking and analysis about the hard instance mining are interesting and the corresponding solution achieves a great improvement of the performance.



Weakness:
- For fair comparison to ST3D, could the authors further provide the solution to combine the proposed method with PV-RCNN  and further show the performance comparison?
- Is the proposed method sensitive to the pseudo-label generation threshold? How to determine the high-pass and low-pass threshold in line 210?
- In line 223, the authors claim that their model achieves good performance even starting with low-quality pseudo labels. Could you please give some analysis and visualization about pseudo label adjustment during the training phase to further illustrate why and how the proposed method could achieve good results with low-quality initialized pseudo labels?

**Time Spent Reviewing:**

2

---

> ### Author Response · Authors · 2021-08-10
> **Our Response to Reviewer DPJL**
>
> We thank the reviewer for the constructive comments.
>
> Q1. To combine the proposed approach with PV-RCNN for a fair comparison with ST3D.
>
> Per the reviewer's request, we build the proposed 3D-CoCo on PV-RCNN (i.e., 3D-CoCo*) to make a fair comparison with ST3D [35]. Due to the time constraint of the rebuttal period, we partly build our method without using the hard sample mining scheme. The experimental result on the nuScenes $\rightarrow$ KITTI benchmark is represented as follows:
>
> | Method | $\text{AP}_\text{BEV}$ (%) | $\text{AP}_\text{3D}$ (%)
> | :-----| ----: | :----:
> | Source Only | 60.48 | 49.47
> | ST3D [35] | 84.29 | 72.94
> | 3D-CoCo* | 85.01 | 74.12
> | Oracle | 88.98 | 82.50
>
> From above, 3D-CoCo is superior to ST3D on the new base detector of PV-RCNN. It is worth noting that the proposed method of hard sample mining has the potential to further improve the adaptation performance. We'll implement the full 3D-CoCo approach based on PV-RCNN in the revision.
>
> &nbsp;
>
> Q2. (1) Sensitivity analyses of 3D-CoCo to the pseudo-label generation thresholds; (2) how to determine the thresholds?
>
> (1) Fig. 4 (c2) provides the sensitivity analyses of the high-pass threshold for generating pseudo-labels (i.e., the confidence score ranging from 0.5 to 0.8). By comparing the results with those of self-training, as shown in Fig. 4 (c1), we can see that:
>
> - 3D-CoCo is robust to different threshold values. It outperforms the Source Only baseline consistently, with all possible values, and significantly, with 50%-100% final performance gains in $\text{AP}_\text{3D}$. In contrast, self-training fails to improve the Source Only baseline with some threshold values.
>
> - At different thresholds, the performance of 3D-CoCo is stably increased throughout the training procedure, while the training process of self-training is quite unstable.
>
> (2) We adopt the value of the low-pass threshold (0.2) from the existing work of ST3D [35], because the empirical evidence given by ST3D shows that the sensitivity of the detection model to the low-pass threshold is lower than the high-pass threshold.
>
> Besides, we follow the hyper-parameter tuning strategy of ST3D and set the high-pass threshold to 0.7 on all benchmarks. We observe that:
>
> - In practice, as mentioned above, our method is more robust to different values of the high-pass threshold than the self-training framework of ST3D.
>
> - We only determined the high-pass threshold on the first benchmark, and found it robust on other benchmarks, in the sense that no further tuning on these hyper-parameters is required.
> &nbsp;
>
> &nbsp;
>
> Q3. Analyses and visualizations about pseudo-label adjustment during the training phase to illustrate why and how the proposed method could achieve good results with low-quality initialized pseudo-labels.
>
> The visualization in Fig. 4 shows that, in 3D-CoCo, the progressive update of pseudo-labels is closely related to the final results, and the key to achieving good results with low-quality initialized pseudo-labels lies in three aspects:
>
> (1) In Fig. 4 (b2), we use the ratio of true positive and false positive predictions to indicate the quality of pseudo-labels.
>
> - At the beginning of the training phase, we have low-quality initialized pseudo-labels due to the large distribution shift across domains.
>
> - At around the 15th training epoch after the warm-up stage (w.r.t. Line 5, Alg. 1), #TPs/#FPs of the blue and green curves grow rapidly, and the model performance in Fig. 4 (b1) grows simultaneously. During these periods, we adjust pseudo-labels in each training epoch (Line 7, Alg. 1).
>
> Analyses: In 3D-CoCo, the iterative training scheme of (i) pseudo-label adjustment and (ii) model optimization gradually improves the quality of pseudo-labels, which has great effects on the domain adaptation performance.
>
> (2) In Fig. 4 (a2), we validate the effectiveness of the contrastive loss by only using the low-quality initialized pseudo-labels throughout the training phase, without pseudo-label adjustment. We can see that, compared with the self-training method, #TPs/#FPs of 3D-CoCo increases more rapidly during the training phase, and the model performance in Fig. 4 (a1) also grows remarkably.
>
> Analyses: The contrastive loss contributes to adapting the feature distribution of the target domain to the source domain. Even without high-quality pseudo-labels of the target data, it improves the detection results of target samples under the guidance of source labeled data.
>
> (3) Also shown in Fig. 4 (b1-b2), the proposed method of hard sample mining (HSM) further promotes the pseudo-label adjustment in the training phase, thus improving the domain adaptation performance.

---

### Official Review · Reviewer_qi8q · 2021-07-21

**Rating:** 7
**Confidence:** 4

**Summary:**

This work presents an unsupervised domain adaptation for point cloud detection in the autonomous driving scenario, which employs a domain-specific 3D encoder for domain-specific feature extraction, a shared BEV feature transformation module for domain-invariant instance-level feature alignment. It adapts the contrastive instance alignment with geometry priors to align the BEV feature distributions induced by pseudo-labels and the true distribution. The proposed 3D-CoCo works on three transferring settings, effectively closing the domain gaps and can outperform recent SOTA methods.

**Limitations And Societal Impact:**

It has included the limitations. The potential negative societal impact of their work  may not be available.

**Main Review:**

This study is useful for researchers in both fields of unsupervised domain transfer and point cloud detection in the autonomous driving scenario. The proposed 3D-CoCo tries to utilize the domain knowledge induced by this special task to produce an effective unsupervised domain adaptation, rather than simply adopts the tricks used in 2D unsupervised domain adaptation. To be specific, it is nice to see that 3D-CoCo would like to specifically handle the domain drift caused by geometric variations, and designs a hard negative mining strategy by examining the statistics of pseudo-label quality. The experiments show good results on three datasets collected by heterogeneous LiDAR sensors, and outperform recent UDA methods on point cloud detection.

But the presentation needs to be polished. At first, symbols and terms in equations (4) and (5) may require a careful definition and remove some careless misuses. And the functionalities of the introduced terms and symbols should be briefly discussed. The authors claimed the effectiveness of co-training beyond self-training, but how they are especially different, or in what aspects co-training is better should be explicitly stated.

Overall, this paper is well written, clearly illustrated, and appropriately structured.

**Time Spent Reviewing:**

2.5 hours

---

> ### Author Response · Authors · 2021-08-10
> **Our Response to Reviewer qi8q**
>
> Thank you very much for the encouraging and constructive comments.
>
> Q1. Symbols and terms in equations (4) and (5).
>
> Eq. (4) searches for the positive sample pairs and Eq. (5) uses them to optimize the contrastive loss. Below we clarify the symbols and terms in these equations. **Please use the Safari or Firefox browser to display the equations in LaTeX properly.**
>
> - $I_i^S$ ($I_j^T$): a sample of the source (target) domain.
>
> - $N_c^S$ ($N_c^T$): the total number of samples of the source (target) domain from the category $c$.
>
> - $|C|$: the number of categories.
>
> - $\Phi(\mathbf{x},\mathbf{y}) = \frac{\mathbf{x}\cdot\mathbf{y}}{\left \|\mathbf{x}\right\|\left \|\mathbf{y}\right\|}$: the cosine distance between the features of $I_i^S$ and $I_{j}^T$.
>
> In Eq. (4), we have $j^\star = \mathop{\mathbf{P}^c(i)} = \mathop{\arg\min}\limits_{1\leq j\leq N_c^T}\{\Phi(I_i^S, I_j^T)\}, ~~ 1\leq i\leq N_c^S, ~~ c=1, 2, \ldots, |C|$. For each $I_i^S$, it finds a positive sample $I_{j^\star}^T$ from the target domain under the same category of $I_i^S$ (indicated by the same pseudo-labels).
>
> As for Eq. (5), we correct the typos in the manuscript and re-write the three loss terms as:
>
> $\mathcal L_\text{intra-class}(S,T) = \sum_{c=1}^{|C|}\sum_{i\in N_c^S}\log\frac{\exp(I^S_i\cdot I^T_{j^\star}/\tau)}{\exp(I_i^S\cdot I_{j^\star}/\tau) + \sum_{j\in N_{\tilde{c}}^T} \exp(I_i^S\cdot I_j^T)},$
>
> $\mathcal L_\text{inter-class}(S) = \sum_{c=1}^{|C|}\sum_{i\in N_c^S, j\in N_c^S, i\neq j}\log\frac{\exp(I^S_i\cdot I^S_j/\tau)}{\exp(I_i^S\cdot I_j^S/\tau) + \sum_{j\in N_{\tilde{c}}^{S}} \exp(I_i^S\cdot I_j^S)},$
>
> $\mathcal L_\text{inter-class}(T) = \sum_{c=1}^{|C|}\sum_{i\in N_c^T, j\in N_c^T, i\neq j}\log\frac{\exp(I^T_i\cdot I^T_j/\tau)}{\exp(I_i^T\cdot I_j^T/\tau) + \sum_{j\in N_{\tilde{c}}^{T}} \exp(I_i^T\cdot I_j^T)}.$
>
> The loss terms have two parts:
>
> - $\mathcal L_\text{intra-class}(S,T)$ encourages to close the intra-class distances of samples in the same category cross domains. Note that $j^\star=\mathbf{P}^c(i)$ is derived from Eq. (4) and $\tilde{c}=|C|\setminus c$ denotes all rest categories except that of the positive pair $(I_i^S, I_{j^\star}^T)$. All samples in $\tilde{c}$ are considered as negative samples.
>
> - $\mathcal L_\text{inter-class}(S)$ and $\mathcal L_\text{inter-class}(T)$ encourage to close the inter-class distances of samples in the same category within same domain. Like $\mathcal L_\text{intra-class}(S,T)$, samples in the positive pairs are from the same category and those in the negative pairs are from different categories.
>
> &nbsp;
>
> Q2. Co-training vs. self-training.
>
> In addition to the comparisons in Lines 294-298 in the manuscript, here we summarize the differences between co-training and self-training planned to be included in the revision.
>
> Self-training (typically for single-domain semi-supervised learning):
>
> - A teacher model is trained on the labeled data.
>
> - The teacher model generates pseudo-labels on unlabeled data.
>
> - A student model is trained to optimize the loss on human labels and pseudo-labels jointly.
>
> Co-training (for cross-domain transfer learning, also shown in Alg. 1):
>
> - A source model is trained on the labeled data of the source domain.
>
> - A target model initialized with the source model generates (updates) pseudo-labels for unlabeled data of the target domain.
>
> - The pseudo-labels are then used to mine the hard samples to augment the target domain.
>
> - The target model and the source model are co-trained with the detection loss (for source data), a regularization term (for target data), as well as the contrastive loss (for both).
>
> - Go back to the second step and then iterate.
>
> The most essential difference between the two strategies is whether the source (teacher) model and the target (student) model are jointly learned. The contrastive loss transfers knowledge from the source domain by narrowing the distance between the sample features of different domains under the same category (indicated by pseudo-labels), thus facilitating the learning process on the unlabeled target set. As illustrated by the empirical results in Fig. 4 and the analyses of error bars in the supplementary material, in contrast with self-training, the proposed co-training strategy not only improves the accuracy but also performs more stably.

---

### Decision · Program_Chairs · 2021-09-27

**Decision:**

Accept (Poster)

**Comment:**

This work addresses a practically important domain adaptation problem in 3D point cloud detection. The authors establish a sound perspective to this specific problem, and propose a new and effective framework that is not simply an extension of existing UDA algorithms to point cloud detection. The authors also construct new datasets to promote future research on this topic.

All reviewers agree on the paper’s main contributions. They also recommend further polishing of the paper writing. AC agrees with the reviewers, and recommend acceptance of the paper.